# Bayesian Optimization for Probabilistic Programs

**Tom Rainforth**[†]    **Tuan Anh Le**[†]    **Jan-Willem van de Meent**[‡]
**Michael A. Osborne**[†]    **Frank Wood**[†]
[†] Department of Engineering Science, University of Oxford
[‡] College of Computer and Information Science, Northeastern University
{twgr,tuananh,mosb,fwood}@robots.ox.ac.uk, j.vandemeent@northeastern.edu

## Abstract

We present the first general purpose framework for marginal maximum a posteriori estimation of probabilistic program variables. By using a series of code transformations, the evidence of any probabilistic program, and therefore of any graphical model, can be optimized with respect to an arbitrary subset of its sampled variables. To carry out this optimization, we develop the first Bayesian optimization package to directly exploit the source code of its target, leading to innovations in problem-independent hyperpriors, unbounded optimization, and implicit constraint satisfaction; delivering significant performance improvements over prominent existing packages. We present applications of our method to a number of tasks including engineering design and parameter optimization.

## 1   Introduction

Probabilistic programming systems (PPS) allow probabilistic models to be represented in the form of a generative model and statements for conditioning on data [4, 9, 10, 16, 17, 29]. Their core philosophy is to decouple model specification and inference, the former corresponding to the user-specified program code and the latter to an inference engine capable of operating on arbitrary programs. Removing the need for users to write inference algorithms significantly reduces the burden of developing new models and makes effective statistical methods accessible to non-experts.

Although significant progress has been made on the problem of general purpose *inference* of program variables, less attention has been given to their *optimization*. Optimization is an essential tool for effective machine learning, necessary when the user requires a single estimate. It also often forms a tractable alternative when full inference is infeasible [18]. Moreover, coincident optimization and inference is often required, corresponding to a marginal maximum a posteriori (MMAP) setting where one wishes to maximize some variables, while marginalizing out others. Examples of MMAP problems include hyperparameter optimization, expectation maximization, and policy search [27].

In this paper we develop the first system that extends probabilistic programming (PP) to this more general MMAP framework, wherein the user specifies a model in the same manner as existing systems, but then selects some subset of the sampled variables in the program to be optimized, with the rest marginalized out using existing inference algorithms. The *optimization query* we introduce can be implemented and utilized in any PPS that supports an inference method returning a marginal likelihood estimate. This framework increases the scope of models that can be expressed in PPS and gives additional flexibility in the outputs a user can request from the program.

MMAP estimation is difficult as it corresponds to the optimization of an intractable integral, such that the optimization target is expensive to evaluate and gives noisy results. Current PPS inference engines are typically unsuited to such settings. We therefore introduce BOPP[1] (Bayesian optimization for probabilistic programs) which couples existing inference algorithms from PPS, like *Anglican* [29], with a new Gaussian process (GP) [22] based Bayesian optimization (BO) [11, 15, 20, 23] package.

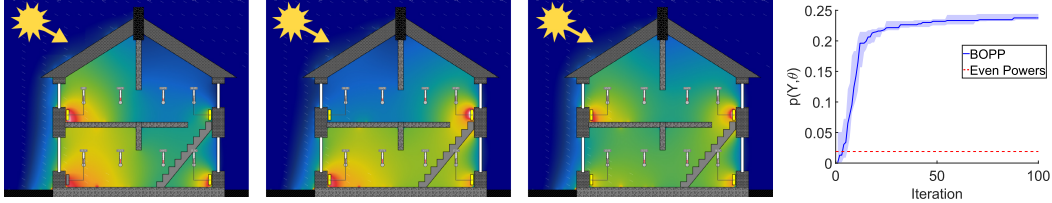

Figure 1: Simulation-based optimization of radiator powers subject to varying solar intensity. Shown are output heat maps from Energy2D [30] simulations at one intensity, corresponding from left to right to setting all the radiators to the same power, the best result from a set of randomly chosen powers, and the best setup found after 100 iterations of BOPP. The far right plot shows convergence of the evidence of the respective model, giving the median and 25/75% quartiles.

```
(defopt house-heating [alphas] [powers]
  (let [solar-intensity (sample weather-prior)
        powers (sample (dirichlet alphas))
        temperatures (simulate solar-intensity powers)]
    (observe abc-likelihood temperatures)))
```

Figure 2: BOPP query for optimizing the power allocation to radiators in a house. Here `weather-prior` is a distribution over the solar intensity and a uniform Dirichlet prior with concentration `alpha` is placed over the powers. Calling **simulate** performs an Energy2D simulation of house temperatures. The utility of the resulting output is conditioned upon using **abc-likelihood**. Calling **doopt** on this query invokes the BOPP algorithm to perform MMAP estimation, where the second input `powers` indicates the variable to be optimized.

To demonstrate the functionality provided by BOPP, we consider an example application of engineering design. Engineering design relies extensively on simulations which typically have two things in common: the desire of the user to find a single best design and an uncertainty in the environment in which the designed component will live. Even when these simulations are deterministic, this is an approximation to a truly stochastic world. By expressing the utility of a particular design-environment combination using an approximate Bayesian computation (ABC) likelihood [5], one can pose this as a MMAP problem, optimizing the design while marginalizing out the environmental uncertainty.

Figure 1 illustrates how BOPP can be applied to engineering design, taking the example of optimizing the distribution of power between radiators in a house so as to homogenize the temperature, while marginalizing out possible weather conditions and subject to a total energy budget. The probabilistic program shown in Figure 2 allows us to define a prior over the uncertain weather, while conditioning on the output of a deterministic simulator (here Energy2D [30]-a finite element package for heat transfer) using an ABC likelihood. BOPP now allows the required coincident inference and optimization to be carried out automatically, directly returning increasingly optimal configurations.

BO is an attractive choice for the required optimization in MMAP as it is typically efficient in the number of target evaluations, operates on non-differentiable targets, and incorporates noise in the target function evaluations. However, applying BO to probabilistic programs presents challenges, such as the need to give robust performance on a wide range of problems with varying scaling and potentially unbounded support. Furthermore, the target program may contain unknown constraints, implicitly defined by the generative model, and variables whose type is unknown (i.e. they may be continuous or discrete).

On the other hand, the availability of the target source code in a PPS presents opportunities to overcome these issues and go beyond what can be done with existing BO packages. BOPP exploits the source code in a number of ways, such as optimizing the acquisition function using the original generative model to ensure the solution satisfies the implicit constraints, performing adaptive domain scaling to ensure that GP kernel hyperparameters can be set according to problem-independent hyperpriors, and defining an adaptive non-stationary mean function to support unbounded BO.

Together, these innovations mean that BOPP can be run in a manner that is fully black-box from the user's perspective, requiring only the identification of the target variables relative to current syntax for operating on arbitrary programs. We further show that BOPP is competitive with existing BO engines for direct optimization on common benchmarks problems that do not require marginalization.

## 2 Background

### 2.1 Probabilistic Programming

Probabilistic programming systems allow users to define probabilistic models using a domain-specific programming language. A probabilistic program implicitly defines a distribution on random variables, whilst the system back-end implements general-purpose inference methods.

PPS such as Infer.Net [17] and Stan [4] can be thought of as defining graphical models or factor graphs. Our focus will instead be on systems such as Church [9], Venture [16], WebPPL [10], and Anglican [29], which employ a general-purpose programming language for model specification. In these systems, the set of random variables is dynamically typed, such that it is possible to write programs in which this set differs from execution to execution. This allows an unspecified number of random variables and incorporation of arbitrary black box deterministic functions, such as was exploited by the `simulate` function in Figure 2. The price for this expressivity is that inference methods must be formulated in such a manner that they are applicable to models where the density function is intractable and can only be evaluated during forwards simulation of the program.

One such general purpose system, *Anglican*, will be used as a reference in this paper. In Anglican, models are defined using the inference macro `defquery`. These models, which we refer to as queries [9], specify a joint distribution $p(Y, X)$ over data $Y$ and variables $X$. Inference on the model is performed using the macro `doquery`, which produces a sequence of approximate samples from the conditional distribution $p(X|Y)$ and, for importance sampling based inference algorithms (e.g. sequential Monte Carlo), a marginal likelihood estimate $p(Y)$.

Random variables in an Anglican program are specified using `sample` statements, which can be thought of as terms in the prior. Conditioning is specified using `observe` statements which can be thought of as likelihood terms. Outputs of the program, taking the form of posterior samples, are indicated by the return values. There is a finite set of `sample` and `observe` statements in a program source code, but the number of times each statement is called can vary between executions. We refer the reader to `http://www.robots.ox.ac.uk/˜fwood/anglican/` for more details.

### 2.2 Bayesian Optimization

Consider an arbitrary black-box target function $f : \vartheta \to \mathbb{R}$ that can be evaluated for an arbitrary point $\theta \in \vartheta$ to produce, potentially noisy, outputs $\hat{w} \in \mathbb{R}$. BO [15, 20] aims to find the global maximum

$$\theta^* = \operatorname*{argmax}_{\theta \in \vartheta} f(\theta). \tag{1}$$

The key idea of BO is to place a prior on $f$ that expresses belief about the space of functions within which $f$ might live. When the function is evaluated, the resultant information is incorporated by conditioning upon the observed data to give a posterior over functions. This allows estimation of the expected value and uncertainty in $f(\theta)$ for all $\theta \in \vartheta$. From this, an acquisition function $\zeta : \vartheta \to \mathbb{R}$ is defined, which assigns an expected utility to evaluating $f$ at particular $\theta$, based on the trade-off between exploration and exploitation in finding the maximum. When direct evaluation of $f$ is expensive, the acquisition function constitutes a cheaper to evaluate substitute, which is optimized to ascertain the next point at which the target function should be evaluated in a sequential fashion. By interleaving optimization of the acquisition function, evaluating $f$ at the suggested point, and updating the surrogate, BO forms a global optimization algorithm that is typically very efficient in the required number of function evaluations, whilst naturally dealing with noise in the outputs. Although alternatives such as random forests [3, 14] or neural networks [26] exist, the most common prior used for $f$ is a GP [22]. For further information on BO we refer the reader to the recent review by Shahriari et al [24].

## 3 Problem Formulation

Given a program defining the joint density $p(Y, X, \theta)$ with fixed $Y$, our aim is to optimize with respect to a subset of the variables $\theta$ whilst marginalizing out latent variables $X$

$$\theta^* = \operatorname*{argmax}_{\theta \in \vartheta} p(\theta|Y) = \operatorname*{argmax}_{\theta \in \vartheta} p(Y, \theta) = \operatorname*{argmax}_{\theta \in \vartheta} \int p(Y, X, \theta) dX. \tag{2}$$

To provide syntax to differentiate between $\theta$ and $X$, we introduce a new query macro `defopt`. The syntax of `defopt` is identical to `defquery` except that it has an additional input identifying the variables to be optimized. To allow for the interleaving of inference and optimization required in MMAP estimation, we further introduce `doopt`, which, analogous to `doquery`, returns a lazy sequence $\{\hat{\theta}_m^*, \hat{\Omega}_m^*, \hat{u}_m^*\}_{m=1,\dots}$ where $\hat{\Omega}_m^* \subseteq X$ are the program outputs associated with $\theta = \hat{\theta}_m^*$ and each $\hat{u}_m^* \in \mathbb{R}^+$ is an estimate of the corresponding log marginal $\log p(Y, \hat{\theta}_m^*)$ (see Section 4.2). The sequence is defined such that, at any time, $\hat{\theta}_m^*$ corresponds to the point expected to be most optimal of those evaluated so far and allows both inference and optimization to be carried out online.

Although no restrictions are placed on $X$, it is necessary to place some restrictions on how programs use the optimization variables $\theta = \phi_{1:K}$ specified by the optimization argument list of `defopt`. First, each optimization variable $\phi_k$ must be bound to a value directly by a `sample` statement with fixed measure-type distribution argument. This avoids change of variable complications arising from nonlinear deterministic mappings. Second, in order for the optimization to be well defined, the program must be written such that any possible execution trace binds each optimization variable $\phi_k$ exactly once. Finally, although any $\phi_k$ may be lexically multiply bound, it must have the same base measure in all possible execution traces, because, for instance, if the base measure of a $\phi_k$ were to change from Lebesgue to counting, the notion of optimality would no longer admit a conventional interpretation. Note that although the transformation implementations shown in Figure 3 do not contain runtime exception generators that disallow continued execution of programs that violate these constraints, those actually implemented in the BOPP system do.

## 4   Bayesian Program Optimization

In addition to the syntax introduced in the previous section, there are five main components to BOPP:

- A program transformation, q→q-marg, allowing estimation of the evidence $p(Y, \theta)$ at a fixed $\theta$.
- A high-performance, GP based, BO implementation for actively sampling $\theta$.
- A program transformation, q→q-prior, used for automatic and adaptive domain scaling, such that a problem-independent hyperprior can be placed over the GP hyperparameters.
- An adaptive non-stationary mean function to support unbounded optimization.
- A program transformation, q→q-acq, and annealing maximum likelihood estimation method to optimize the acquisition function subject the implicit constraints imposed by the generative model.

Together these allow BOPP to perform online MMAP estimation for arbitrary programs in a manner that is black-box from the user's perspective - requiring only the definition of the target program in the same way as existing PPS and identifying which variables to optimize. The BO component of BOPP is both probabilistic programming and language independent, and is provided as a stand-alone package.[2] It requires as input only a target function, a sampler to establish rough input scaling, and a problem specific optimizer for the acquisition function that imposes the problem constraints.

Figure 3 provides a high level overview of the algorithm invoked when `doopt` is called on a query q that defines a distribution $p(Y, a, \theta, b)$. We wish to optimize $\theta$ whilst marginalizing out $a$ and $b$, as indicated by the the second input to q. In summary, BOPP performs iterative optimization in 5 steps

- Step 1 (blue arrows) generates unweighted samples from the transformed prior program q-prior (*top center*), constructed by removing all conditioning. This initializes the domain scaling for $\theta$.
- Step 2 (red arrows) evaluates the marginal $p(Y, \theta)$ at a small number of the generated $\hat{\theta}$ by performing inference on the marginal program q-marg (*middle centre*), which returns samples from the distribution $p(a, b|Y, \theta)$ along with an estimate of $p(Y, \theta)$. The evaluated points (*middle right*) provide an initial domain scaling of the outputs and starting points for the BO surrogate.
- Step 3 (black arrow) fits a mixture of GPs posterior [22] to the scaled data (*bottom centre*) using a problem independent hyperprior. The solid blue line and shaded area show the posterior mean and $\pm 2$ standard deviations respectively. The new estimate of the optimum $\hat{\theta}^*$ is the value for which the mean estimate is largest, with $\hat{u}^*$ equal to the corresponding mean value.

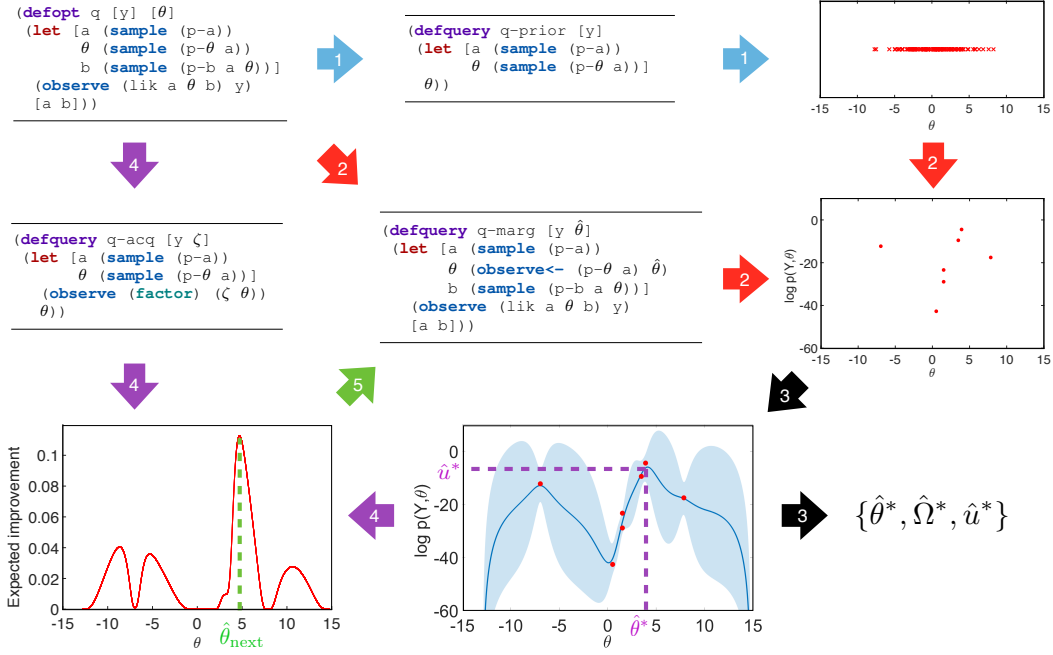

Figure 3: Overview of the BOPP algorithm, description given in main text. p-a, p-$\theta$, p-b and lik all represent distribution object constructors. **factor** is a special distribution constructor that assigns probability $p(y) = y$, in this case $y = \zeta(\theta)$.

- Step 4 (purple arrows) constructs an acquisition function $\zeta \colon \vartheta \to \mathbb{R}^+$ (*bottom left*) using the GP posterior. This is optimized, giving the next point to evaluate $\hat{\theta}_{\text{next}}$, by performing annealed importance sampling on a transformed program q-acq (*middle left*) in which all **observe** statements are removed and replaced with a single **observe** assigning probability $\zeta(\theta)$ to the execution.

- Step 5 (green arrow) evaluates $\hat{\theta}_{\text{next}}$ using q-marg and continues to step 3.

### 4.1 Program Transformation to Generate the Target

Consider the **defopt** query q in Figure 3, the body of which defines the joint distribution $p(Y, a, \theta, b)$. Calculating (2) (defining $X = \{a, b\}$) using a standard optimization scheme presents two issues: $\theta$ is a random variable within the program rather than something we control and its probability distribution is only defined conditioned on $a$.

We deal with both these issues simultaneously using a program transformation similar to the disintegration transformation in Hakaru [31]. Our *marginal* transformation returns a new **query** object, q-marg as shown in Figure 3, that defines the same joint distribution on program variables and inputs, but now accepts the value for $\theta$ as an input. This is done by replacing all **sample** statements associated with $\theta$ with equivalent **observe<-** statements, taking $\theta$ as the observed value, where **observe<-** is identical to **observe** except that it returns the observed value. As both **sample** and **observe** operate on the same variable type - a distribution object - this transformation can always be made, while the identical returns of **sample** and **observe<-** trivially ensures validity of the transformed program.

### 4.2 Bayesian Optimization of the Marginal

The target function for our BO scheme is $\log p(Y, \theta)$, noting $\operatorname{argmax} f(\theta) = \operatorname{argmax} \log f(\theta)$ for any $f \colon \vartheta \to \mathbb{R}^+$. The log is taken because GPs have unbounded support, while $p(Y, \theta)$ is always positive, and because we expect variations over many orders of magnitude. PPS with importance sampling based inference engines, e.g. sequential Monte Carlo [29] or the particle cascade [21], can return noisy estimates of this target given the transformed program q-marg.

Our BO scheme uses a GP prior and a Gaussian likelihood. Though the rationale for the latter is predominantly computational, giving an analytic posterior, there are also theoretical results suggesting that this choice is appropriate [2]. We use as a default covariance function a combination of a Matérn-3/2 and Matérn-5/2 kernel. By using automatic domain scaling as described in the next section, problem independent priors are placed over the GP hyperparameters such as the length scales and observation noise. Inference over hyperparameters is performed using Hamiltonian Monte Carlo (HMC) [6], giving an unweighted mixture of GPs. Each term in this mixture has an analytic distribution fully specified by its mean function $\mu_m^i \colon \vartheta \to \mathbb{R}$ and covariance function $k_m^i \colon \vartheta \times \vartheta \to \mathbb{R}$, where $m$ indexes the BO iteration and $i$ the hyperparameter sample.

This posterior is first used to estimate which of the previously evaluated $\hat{\theta}_j$ is the most optimal, by taking the point with highest expected value , $\hat{u}_m^* = \max_{j \in 1...m} \sum_{i=1}^N \mu_m^i(\hat{\theta}_j)$. This completes the definition of the output sequence returned by the `doopt` macro. Note that as the posterior updates globally with each new observation, the relative estimated optimality of previously evaluated points changes at each iteration. Secondly it is used to define the acquisition function $\zeta$, for which we take the expected improvement [25], defining $\sigma_m^i(\theta) = \sqrt{k_m^i(\theta, \theta)}$ and $\gamma_m^i(\theta) = \frac{\mu_m^i(\theta) - \hat{u}_m^*}{\sigma_m^i(\theta)}$,

$$\zeta(\theta) = \sum_{i=1}^N \left( \mu_m^i(\theta) - \hat{u}_m^* \right) \Phi\left(\gamma_m^i(\theta)\right) + \sigma_m^i(\theta) \phi\left(\gamma_m^i(\theta)\right) \tag{3}$$

where $\phi$ and $\Phi$ represent the pdf and cdf of a unit normal distribution respectively. We note that more powerful, but more involved, acquisition functions, e.g. [12], could be used instead.

## 4.3 Automatic and Adaptive Domain Scaling

Domain scaling, by mapping to a common space, is crucial for BOPP to operate in the required black-box fashion as it allows a general purpose and problem independent hyperprior to be placed on the GP hyperparameters. BOPP therefore employs an affine scaling to a $[-1, 1]$ hypercube for both the inputs and outputs of the GP. To initialize scaling for the input variables, we sample directly from the generative model defined by the program. This is achieved using a second transformed program, q-prior, which removes all conditioning, i.e. `observe` statements, and returns $\theta$. This transformation also introduces code to terminate execution of the query once all $\theta$ are sampled, in order to avoid unnecessary computation. As `observe` statements return `nil`, this transformation trivially preserves the generative model of the program, but the probability of the execution changes. Simulating from the generative model does not require inference or calling potentially expensive likelihood functions and is therefore computationally inexpensive. By running inference on q-marg given a small number of these samples as arguments, a rough initial characterization of output scaling can also be achieved. If points are observed that fall outside the hypercube under the initial scaling, the domain scaling is appropriately updated[3] so that the target for the GP remains the $[-1, 1]$ hypercube.

## 4.4 Unbounded Bayesian Optimization via Non-Stationary Mean Function Adaptation

Unlike standard BO implementations, BOPP is not provided with external constraints and we therefore develop a scheme for operating on targets with potentially unbounded support. Our method exploits the knowledge that the target function is a probability density, implying that the area that must be searched in practice to find the optimum is finite, by defining a non-stationary prior mean function. This takes the form of a bump function that is constant within a region of interest, but decays rapidly outside. Specifically we define this bump function in the transformed space as

$$\mu_{\text{prior}}(r; r_e, r_\infty) = \begin{cases} 0 & \text{if } r \leq r_{\text{e}} \\ \log\left(\frac{r - r_{\text{e}}}{r_\infty - r_{\text{e}}}\right) + \frac{r - r_{\text{e}}}{r_\infty - r_{\text{e}}} & \text{otherwise} \end{cases} \tag{4}$$

where $r$ is the radius from the origin, $r_e$ is the maximum radius of any point generated in the initial scaling or subsequent evaluations, and $r_\infty$ is a parameter set to $1.5 r_e$ by default. Consequently, the acquisition function also decays and new points are never suggested arbitrarily far away. Adaptation

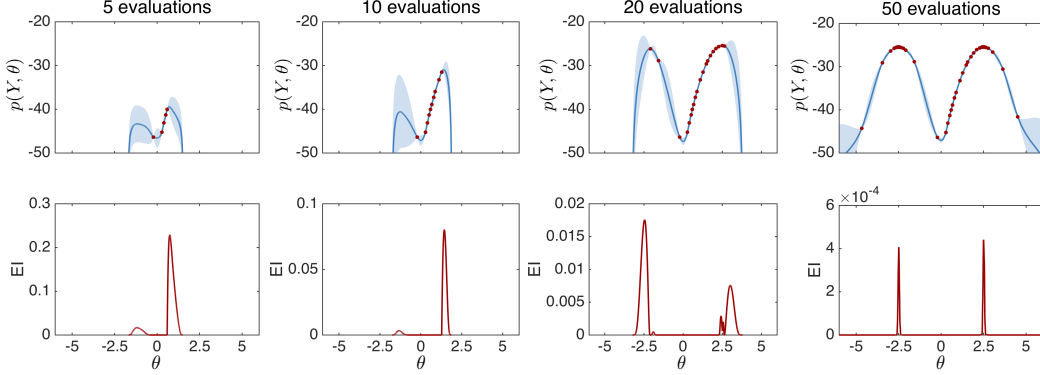

Figure 4: Convergence on an unconstrained bimodal problem with $p(\theta) = \text{Normal}(0, 0.5)$ and $p(Y|\theta) = \text{Normal}(5 - |\theta|, 0.5)$ giving significant prior misspecification. The top plots show a regressed GP, with the solid line corresponding to the mean and the shading shows $\pm 2$ standard deviations. The bottom plots show the corresponding acquisition functions.

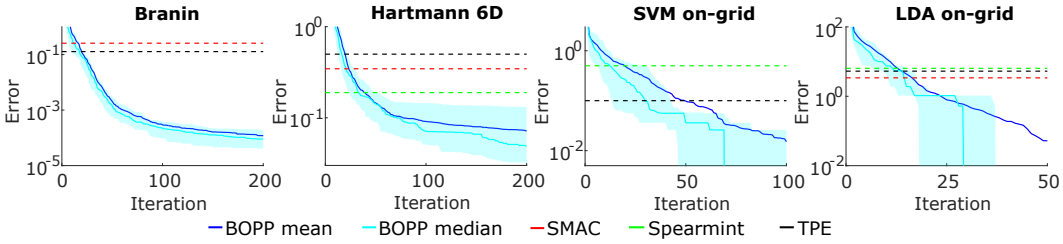

Figure 5: Comparison of BOPP used as an optimizer to prominent BO packages on common benchmark problems. The dashed lines shows the final mean error of SMAC (red), Spearmint (green) and TPE (black) as quoted by [7]. The dark blue line shows the mean error for BOPP averaged over 100 runs, whilst the median and 25/75% percentiles are shown in cyan. Results for Spearmint on Branin and SMAC on SVM on-grid are omitted because both BOPP and the respective algorithms averaged zero error to the provided number of significant figures in [7].

of the scaling will automatically update this mean function appropriately, learning a region of interest that matches that of the true problem, without complicating the optimization by over-extending this region. We note that our method shares similarity with the recent work of Shahriari et al [23], but overcomes the sensitivity of their method upon a user-specified bounding box representing soft constraints, by initializing automatically and adapting as more data is observed.

### 4.5 Optimizing the Acquisition Function

Optimizing the acquisition function for BOPP presents the issue that the query contains implicit constraints that are unknown to the surrogate function. The problem of unknown constraints has been previously covered in the literature [8, 13] by assuming that constraints take the form of a black-box function which is modeled with a second surrogate function and must be evaluated in guess-and-check strategy to establish whether a point is valid. Along with the potentially significant expense such a method incurs, this approach is inappropriate for *equality* constraints or when the target variables are potentially discrete. For example, the Dirichlet distribution in Figure 2 introduces an equality constraint on `powers`, namely that its components must sum to 1.

We therefore take an alternative approach based on directly using the program to optimize the acquisition function. To do so we consider a transformed program `q-acq` that is identical to `q-prior` (see Section 4.3), but adds an additional **observe** statement that assigns a weight $\zeta(\theta)$ to the execution. By setting $\zeta(\theta)$ to the acquisition function, the maximum likelihood corresponds to the optimum of the acquisition function subject to the implicit program constraints. We obtain a maximum likelihood estimate for `q-acq` using a variant of annealed importance sampling [19] in which lightweight Metropolis Hastings (LMH) [28] with local random-walk moves is used as the base transition kernel.

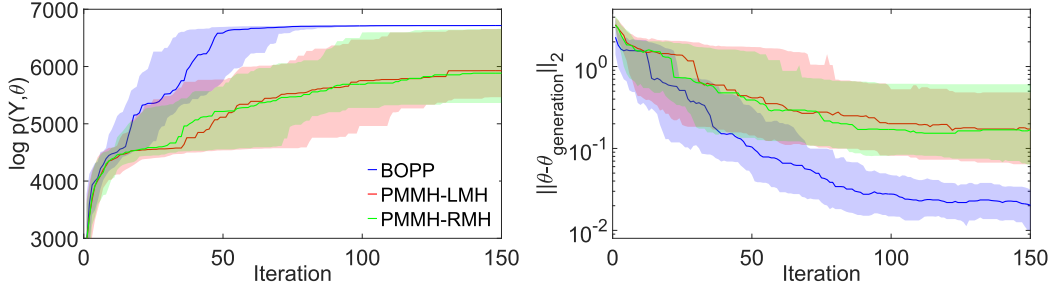

Figure 6: Convergence for transition dynamics parameters of the pickover attractor in terms of the cumulative best $\log p\,(Y, \theta)$ (*left*) and distance to the "true" $\theta$ used in generating the data (*right*). Solid line shows median over 100 runs, whilst the shaded region the 25/75% quantiles.

# 5 Experiments

We first demonstrate the ability of BOPP to carry out unbounded optimization using a 1D problem with a significant prior-posterior mismatch as shown in Figure 4. It shows BOPP adapting to the target and effectively establishing a maxima in the presence of multiple modes. After 20 evaluations the acquisitions begin to explore the right mode, after 50 both modes have been fully uncovered.

Next we compare BOPP to the prominent BO packages SMAC [14], Spearmint [25] and TPE [3] on a number of classical benchmarks as shown in Figure 5. These results demonstrate that BOPP provides substantial advantages over these systems when used simply as an optimizer on both continuous and discrete optimization problems. In particular, it offers a large advantage over SMAC and TPE on the continuous problems (Branin and Hartmann), due to using a more powerful surrogate, and over Spearmint on the others due to not needing to make approximations to deal with discrete problems.

Finally we demonstrate performance of BOPP on a MMAP problem. Comparison here is more difficult due to the dearth of existing alternatives for PPS. In particular, simply running inference on the original query does not return estimates for $p\,(Y, \theta)$. We consider the possible alternative of using our conditional code transformation to design a particle marginal Metropolis Hastings (PMMH, [1]) sampler which operates in a similar fashion to BOPP except that new $\theta$ are chosen using a MH step instead of actively sampling with BO. For these MH steps we consider both LMH [28] with proposals from the prior and the random-walk MH (RMH) variant introduced in Section 4.5. Results for estimating the dynamics parameters of a chaotic pickover attractor, while using an extended Kalman smoother to estimate the latent states are shown in Figure 6. Model details are given in the supplementary material along with additional experiments.

# 6 Discussion and Future Work

We have introduced a new method for carrying out MMAP estimation of probabilistic program variables using Bayesian optimization, representing the first unified framework for optimization and inference of probabilistic programs. By using a series of code transformations, our method allows an arbitrary program to be optimized with respect to a defined subset of its variables, whilst marginalizing out the rest. To carry out the required optimization, we introduce a new GP-based BO package that exploits the availability of the target source code to provide a number of novel features, such as automatic domain scaling and constraint satisfaction.

The concepts we introduce lead directly to a number of extensions of interest, including but not restricted to smart initialization of inference algorithms, adaptive proposals, and nested optimization. Further work might consider maximum marginal likelihood estimation and risk minimization. Though only requiring minor algorithmic changes, these cases require distinct theoretical considerations.

## Acknowledgements

Tom Rainforth is supported by a BP industrial grant. Tuan Anh Le is supported by a Google studentship, project code DF6700. Frank Wood is supported under DARPA PPAML through the U.S. AFRL under Cooperative Agreement FA8750-14-2-0006, Sub Award number 61160290-111668.

## Footnotes

[1] Code available at http://www.github.com/probprog/bopp/

[2]Code available at `http://www.github.com/probprog/deodorant/`

[3]An important exception is that the output mapping to the bottom of the hypercube remains fixed such that low likelihood new points are not incorporated. This ensures stability when considering unbounded problems.

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
