[Supplementary Material]

# Bayesian Optimization for Probabilistic Programs - Supplementary Materials

**Tom Rainforth**[†]    **Tuan Anh Le**[†]    **Jan-Willem van de Meent**[‡]
**Michael A. Osborne**[†]    **Frank Wood**[†]
[†] Department of Engineering Science, University of Oxford
[‡] College of Computer and Information Science, Northeastern University
{twgr,tuananh,mosb,fwood}@robots.ox.ac.uk, j.vandemeent@northeastern.edu

## 1   Program Transformations

In this section we give a more detailed and language specific description of our program transformations, code for which can be found at http://www.github.com/probprog/bopp.

### 1.1   Anglican

Anglican is a probabilistic programming language integrated into Clojure (a dialect of Lisp) and inherits most of the corresponding syntax. Anglican extends Clojure with the special forms **sample** and **observe** [1]. Each random draw in an Anglican program corresponds to a **sample** call, which can be thought of as a term in the prior. Each **observe** statement applies weighting to a program trace and thus constitutes a term in the likelihood. Compilation of an Anglican program, performed by the macro **query**, corresponds to transforming the code into a variant of continuation-passing style (CPS) code, which results in a function that can be executed using a particular inference algorithm.

Anglican program code is represented by a nested list of expressions, symbols, non-literals for contructing data structures (e.g. [...] for vectors), and command dependent literals (e.g. [...] as a second argument of a **let** statement which is used for binding pairs). In order to perform program transformations, we can recursively traverse this nested list which can be thought of as an abstract syntax tree of the program.

Our program transformations also make use of the Anglican forms **store** and **retrieve**. These allow storing any variable in the probabilistic program's execution trace in a state which is passed around during execution and from which we can retrieve these stored values. The core use for this is to allow the outer query to return variables which are only locally scoped.

To allow for the early termination that will be introduced in Section 1.5, it was necessary to add a mechanism for non-local returns to Anglican. Clojure supports non-local returns only through Java exception handling, via the keywords **try throw**, **catch** and **finally**. Unfortunately, these are not currently supported by Anglican and their behaviour is far from ideal for our purposes. In particular, for programs containing nested **try** statements, throwing to a particular **try** in the stack, as opposed to the most recently invoked, is cumbersome and error prone.

We have instead, therefore, added to Anglican a non-local return mechanism based on the Common Lisp control form **catch**/**throw**. This uses a *catch tag* to link each **throw** to a particular **catch**. For example

```
(catch :tag
  (when (> a 0)
    (throw :tag a))
  0)
```

is equivalent to (**max** a 0). More precisely, **throw** has syntax (**throw** tag value) and will cause the **catch** block with the corresponding tag to exit, returning value. If a **throw** goes uncaught, i.e. it is not contained within a **catch** block with a matching tag, a custom Clojure exception is thrown.

## 1.2 Representations in the Main Paper

In the main paper we presented the code transformations as static transformations as shown in Figure 3. Although for simple programs, such as the given example, these transformations can be easily expressed as static transformations, for more complicated programs it would be difficult to actually implement these as purely static generic transformations in a higher-order language. Therefore, even though all the transformations dynamically execute as shown at runtime, in truth, the generated source code for the prior and acquisition transformations varies from what is shown and has been presented this way in the interest of exposition. Our true transformations exploit **store**, **retrieve**, **catch** and **throw** to generate programs that dynamically execute in the same way at run time as the static examples shown, but whose actual source code varies significantly.

## 1.3 Prior Transformation

The prior transformation recursively traverses the program tree and applies two local transformations. Firstly it replaces all **observe** statements by nil. As **observe** statements return nil, this trivially preserves the generative model of the program, but the probability of the execution changes. Secondly, it inspects the binding variables of **let** forms in order to modify the binding expressions for the optimization variables, as specified by the second input of **defopt**, asserting that these are directly bound to a **sample** statement of the form (**sample** dist). The transformation then replaces this expression by one that stores the result of this sample in Anglican's **store** before returning it. Specifically, if the binding variable in question is phi-k, then the original binding expression (**sample** dist) is transformed into

```
(let [value (sample dist)]
  ;; Store the sampled value in Anglican's store
  (store OPTIM-ARGS-KEY
         'phi-k
         value)
  value)
```

After all these local transformation have been made, we wrap the resulting query block in a **do** form and append an expression extracting the optimization variables using Anglican's **retrieve**. This makes the optimization variables the output of the query. Denoting the list of optimization variable symbols from **defopt** as optim-args and the query body after applying all the above location transformations as ..., the prior query becomes

```
(query query-args
  (do
    ...
    (map (fn [x] (retrieve OPTIM-ARGS-KEY x))
      optim-args)))
```

Note that the difference in syntax from Figure 3 in the main paper is because **defquery** is in truth a syntactic sugar allowing users to bind **query** to a variable. As previously stated, **query** is macro that compiles an Anglican program to its CPS transformation. An important subtlety here is that the order of the returned samples is dictated by optim-args and is thus independent of the order in which the variables were actually sampled, ensuring consistent inputs for the BO package.

We additionally add a check (not shown) to ensure that all the optimization variables have been added to the store, and thus sampled during the execution, before returning. This ensures that our assumption that each optimization variable is assigned for each execution trace is satisfied.

## 1.4 Acquisition Transformation

The acquisition transformation is the same as the prior transformation except we append the acquisition function, ACQ-F, to the inputs and then **observe** its application to the optimization variables before returning. The acquisition query is thus

```
(query [query-args ACQ-F]
  (do
    ...
    (let [theta (map (fn [x] (retrieve OPTIM-ARGS-KEY x))
                     optim-args)]
      (observe (factor) (ACQ-F theta))
      theta)))
```

## 1.5 Early Termination

To ensure that `q-prior` and `q-acq` are cheap to evaluate and that the latter does not include unnecessary terms which complicate the optimization, we wish to avoid executing code that is not required for generating the optimization variables. Ideally we would like to directly remove all such redundant code during the transformations. However, doing so in a generic way applicable to all possible programs in a higher order language represents a significant challenge. Therefore, we instead transform to programs with additional early termination statements, triggered when all the optimization variables have been sampled. Provided one is careful to define the optimization variables as early as possible in the program (in most applications, e.g. hyperparameter optimization, they naturally occur at the start of the program), this is typically sufficient to ensure that the minimum possible code is run in practise.

To carry out this early termination, we first wrap the query in a **catch** block with a uniquely generated tag. We then augment the transformation of an optimization variable's binding described in Section 1.3 to check if all optimization variables are already stored, and invoke a **throw** statement with the corresponding tag if so. Specifically we replace relevant binding expressions (**sample** dist) with

```
(let [value (sample dist)]
  ;; Store the sampled value in Anglican's store
  (store OPTIM-ARGS-KEY
         'phi-k
         value)
  ;; Terminate early if all optimization variables are sampled
  (if (= (set (keys (retrieve OPTIM-ARGS-KEY)))
         (set optim-args))
    (throw BOPP-CATCH-TAG prologue-code)
    value))
```

where `prologue-code` refers to one of the following expressions depending on whether it is used for a prior or an acquisition transformation

```
;; Prior query prologue-code
(map (fn [x] (retrieve OPTIM-ARGS-KEY x))
             optim-args)

;; Acquisition query prologue-code
(do
  (let [theta (map (fn [x] (retrieve OPTIM-ARGS-KEY x))
                   optim-args)]
  (observe (factor) (ACQ-F theta))
  theta))
```

We note that valid programs for both `q-prior` and `q-acq` should always terminate via one of these early stopping criteria and therefore never actually reach the appending statements in the **query** blocks shown in Sections 1.3 and 1.4. As such, these are, in practise, only for exposition and error catching.

## 1.6 Marginal/MMAP Transformation

The marginal transformation inspects all **let** binding pairs and if a binding variable `phi-k` is one of the optimization variables, the binding expression (**sample** dist) is transformed to the following

```
(do (observe dist phi-k-hat)
    phi-k-hat)
```

corresponding to the **observe<-** form used in the main paper.

### 1.7 Error Handling

During program transformation stage, we provide three error-handling mechanisms to enforce the restrictions on the probabilistic programs described in Section 3.

1. We inspect **let** binding pairs and throw an error if an optimization variable is bound to anything other than a **sample** statement.

2. We add code that throws a runtime error if any optimization variable is assigned more than once or not at all.

3. We recursively traverse the code and throw a compilation error if **sample** statements of different base measures are assigned to any optimization variable. At present, we also throw an error if the base measure assigned to an optimization variable is unknown, e.g. because the distribution object is from a user defined **defdist** where the user does not provide the required measure type meta-information.

## 2 Gaussian Process Surrogate in Detail

Informally one can think of a Gaussian Process (GP) [2] as being a nonparametric distribution over functions which is fully specified by a mean function $\mu\colon \vartheta \to \mathbb{R}$ and covariance function $k\colon \vartheta\times\vartheta \to \mathbb{R}$, the latter of which must be a bounded (i.e. $k\left(\theta,\theta'\right) < \infty,\ \forall\theta,\theta' \in \vartheta$) and reproducing kernel. We can describe a function $f$ as being distributed according to a GP:

$$f\left(\theta\right) \sim GP\left(\mu\left(\theta\right), k\left(\theta,\theta'\right)\right) \tag{1}$$

which by definition means that the functional evaluations realized at any finite number of sample points is distributed according to a multivariate Gaussian. Note that the inputs to $\mu$ and $k$ need not be numeric and as such a GP can be defined over anything for which kernel can be defined.

An important property of a GP is that it is conjugate with a Gaussian likelihood. Consider pairs of input-output data points $\{\hat{\theta}_j, \hat{w}_j\}_{j=1:m}$, $\hat{W} = \{\hat{w}_j\}_{j=1:m}$, $\hat{\Theta} = \{\hat{\theta}_j\}_{j=1:m}$ and the separable likelihood function

$$p(\hat{W}|\hat{\Theta}, f) = \prod_{j=1}^{m} p(\hat{w}_j|f(\hat{\theta}_j)) = \prod_{j=1}^{m} \frac{1}{\sigma_n\sqrt{2\pi}} \exp\left(-\frac{\left(\hat{w}_j - f(\hat{\theta}_j)\right)^2}{2\sigma_n^2}\right) \tag{2}$$

where $\sigma_n$ is an observation noise. Using a GP prior $f\left(\theta\right) \sim GP(\mu_{\text{prior}}\left(\theta\right), k_{\text{prior}}\left(\theta,\theta\right))$ leads to an analytic GP posterior

$$\mu_{post}\left(\theta\right) = \mu_{\text{prior}}\left(\theta\right) + k_{\text{prior}}\left(\theta,\hat{\Theta}\right)\left[k_{\text{prior}}\left(\hat{\Theta},\hat{\Theta}\right) + \sigma_n^2 I\right]^{-1}\left(\hat{W} - \mu_{\text{prior}}\left(\hat{\Theta}\right)\right) \tag{3}$$

$$k_{post}\left(\theta,\theta'\right) = k_{\text{prior}}\left(\theta,\theta'\right) - k_{\text{prior}}\left(\theta,\hat{\Theta}\right)\left[k_{\text{prior}}\left(\hat{\Theta},\hat{\Theta}\right) + \sigma_n^2 I\right]^{-1} k_{\text{prior}}\left(\hat{\Theta},\theta'\right) \tag{4}$$

and Gaussian predictive distribution

$$w|\theta,\hat{W},\hat{\Theta} \sim \mathcal{N}\left(\mu_{post}\left(\theta\right), k_{post}\left(\theta,\theta\right) + \sigma_n^2 I\right) \tag{5}$$

where we have used the shorthand $k_{\text{prior}}(\hat{\Theta},\hat{\Theta}) = \begin{bmatrix} k_{\text{prior}}(\hat{\theta}_1,\hat{\theta}_1) & k_{\text{prior}}(\hat{\theta}_1,\hat{\theta}_2) & \dots \\ k_{\text{prior}}(\hat{\theta}_2,\hat{\theta}_1) & k_{\text{prior}}(\hat{\theta}_2,\hat{\theta}_2) & \dots \\ \dots & \dots & \dots \end{bmatrix}$ and similarly for $\mu_{\text{prior}}$, $\mu_{\text{post}}$ and $k_{\text{post}}$.

For our model, $\mu_{\text{prior}}$ is given by (4) in the main paper and we take the combination of a Matérn 3/2 and a Matérn 5/2 kernel for the prior covariance. Let $D = \|\theta\|_0$ be the dimensionality of $\theta$ and define

$$d_{3/2}(\theta, \theta') = \sqrt{\sum_{i=1}^{D} \frac{\theta_i - \theta_i'}{\rho_i}} \tag{6a}$$

$$d_{5/2}(\theta, \theta') = \sqrt{\sum_{i=1}^{D} \frac{\theta_i - \theta_i'}{\varrho_i}} \tag{6b}$$

where $i$ indexes a dimension of $\theta$ and $\rho_i$ and $\varrho_i$ are dimension specific length scale hyperparameters. Our prior covariance function is now given by

$$\begin{aligned} k_{\text{prior}}(\theta, \theta') =& \sigma_{3/2}^2 \left(1 + \sqrt{3}d_{3/2}(\theta, \theta')\right) \exp\left(-\sqrt{3}d_{3/2}(\theta, \theta')\right) + \\ & \sigma_{5/2}^2 \left(1 + \sqrt{5}d_{5/2}(\theta, \theta') + \frac{5}{3}(d_{5/2}(\theta, \theta'))^2\right) \exp\left(-\sqrt{5}d_{5/2}(\theta, \theta')\right) \end{aligned} \tag{7}$$

where $\sigma_{3/2}$ and $\sigma_{5/2}$ represent signal standard deviations for the two respective kernels. The full set of GP hyperparameters is defined by $\alpha = \{\sigma_n, \sigma_{3/2}, \sigma_{5/2}, \rho_{i=1:D}, \varrho_{i=1:D}\}$. A key feature of this kernel is that it is only once differentiable and therefore makes relatively weak assumptions about the smoothness of $f$. The ability to include branching in a probabilistic program means that, in some cases, an even less smooth kernel than (7) might be preferable. However, there is clear a trade-off between generality of the associated reproducing kernel Hilbert space and modelling power. Our package provides the ability for the user to provide an alternative covariance function if desired.

As noted by [3], the performance of BO using a single GP posterior is heavily influenced by the choice of these hyperparameters. We therefore exploit the automated domain scaling introduced in Section 4.3 to define a problem independent hyperprior $p(\alpha)$ (using the knowledge that both the input and outputs vary between $\pm 1$) and perform inference to give a mixture of GPs posterior. We define $p(\alpha) = p(\sigma_n)p(\sigma_{3/2})p(\sigma_{5/2}) \prod_{i=1}^{D} p(\rho_i)p(\varrho_i)$ where

$$\log(\sigma_n) \sim \mathcal{N}(-5, 2) \tag{8a}$$

$$\log(\sigma_{3/2}) \sim \mathcal{N}(-7, 0.5) \tag{8b}$$

$$\log(\sigma_{5/2}) \sim \mathcal{N}(-0.5, 0.15) \tag{8c}$$

$$\log(\rho_i) \sim \mathcal{N}(-1.5, 0.5) \quad \forall i \in \{1, \ldots, D\} \tag{8d}$$

$$\log(\varrho_i) \sim \mathcal{N}(-1, 0.5) \quad \forall i \in \{1, \ldots, D\}. \tag{8e}$$

The rationale of this hyperprior is that the smoother Matérn 5/2 kernel should be the dominant effect and model the higher length scale variations. The Matérn 3/2 kernel is included in case the evidence suggests that the target is less smooth than can be modelled with the Matérn 5/2 kernel and to provide modelling of smaller scale variations around the optimum.

To perform the required inference for the GP hyperparameters we use Hamiltonian Monte Carlo [4] (HMC). HMC was chosen because of the availability of analytic derivatives of the GP log marginal likelihoods. As we found that the performance of HMC was often poor unless a good initialization point was used, BOPP runs a small number of independent chains and allocates part of the computational budget to their initialization using a L-BFGS optimizer [5].

## 3 Additional Experiments and Details

### 3.1 Industrial Design

In this case study, illustrated in Figure 1 in the main paper, we optimize the parameters of a stochastic engineering simulation. We use the Energy2D system from [6] to perform finite-difference numerical simulation of the heat equation and Navier-Stokes equations in a user-defined geometry.

In our setup, we designed a 2-dimensional representation of a house with 4 interconnected rooms using the GUI provided by Energy2D. The left side of the house receives morning sun, modelled at a constant incident angle of $30°$. We assume a randomly distributed solar intensity and simulate the heating of a cold house in the morning by 4 radiators, one in each of the rooms. The radiators

are given a fixed budget of total power density $P_{\text{budget}}$. The optimization problem is to distribute this power budget across radiators in a manner that minimizes the variance in temperatures across 8 locations in the house.

Energy2D is written in Java, which allows the simulation to be integrated directly into an Anglican program that defines a prior on model parameters and an ABC likelihood for evaluating the utility of the simulation outputs. Figure 2 in the main paper shows the corresponding program query. In this, we define a Clojure function `simulate` that accepts a solar power intensity $I_{\text{sun}}$ and power densities for the radiators $P_{\text{r}}$, returning the thermometer temperature readings $\{T_{i,t}\}$. We place a symmetric Dirichlet prior on $\frac{P_r}{P_{\text{budget}}}$ and a gamma prior on $\frac{I_{\text{sun}}}{I_{base}}$, where $P_{\text{budget}}$ and $I_{base}$ are constants. This gives the generative model:

$$p_r \sim \text{Dirichlet}([1,1,1,1]) \tag{9}$$
$$P_r \leftarrow P_{\text{budget}} \cdot p_r \tag{10}$$
$$\upsilon \sim \text{Gamma}(5,1) \tag{11}$$
$$I_{\text{sun}} \leftarrow I_{\text{base}} \cdot \upsilon. \tag{12}$$

After using these to call `simulate`, the standard deviations of the returned temperatures is calculated for each time point,

$$\omega_t = \sqrt{\sum_{i=1}^{8} T_{i,t}^2 - \left(\sum_{i=1}^{8} T_{i,t}\right)^2} \tag{13}$$

and used in the ABC likelihood `abc-likelihood` to weight the execution trace using a multivariate Gaussian:

$$p\left(\{T_{i,t}\}_{i=1:8,t=1:\tau}\right) = \text{Normal}\left(\omega_{t=1:\tau}; \mathbf{0}, \sigma_T^2 \mathbf{I}\right)$$

where $\mathbf{I}$ is the identity matrix and $\sigma_T = 0.8^{\circ}\text{C}$ is the observation standard deviation.

Figure 1 in the main paper demonstrates the improvement in homogeneity of temperatures as a function of total number of simulation evaluations. Visual inspection of the heat distributions also shown in Figure 1 confirms this result, which serves as an exemplar of how BOPP can be used to estimate marginally optimal simulation parameters.

## 3.2 Hyperparameter Optimization

We next consider an illustrative case study of optimizing the hyperparameters in a multivariate Gaussian mixture model. We consider a Bayesian formulation with a symmetric Dirichlet prior on the mixture weights and a Gaussian-inverse-Wishart prior on the likelihood parameters:

$$\boldsymbol{\pi} \sim \text{Dir}(\alpha, \ldots, \alpha) \tag{14}$$
$$(\boldsymbol{\mu}_k, \boldsymbol{\Sigma}_k) \sim \text{NIW}(\boldsymbol{\mu}_0, \kappa, \boldsymbol{\Psi}, \nu) \qquad \text{for } k = 1, \ldots, K \tag{15}$$
$$z_n \sim \text{Disc}(\boldsymbol{\pi}) \tag{16}$$
$$\boldsymbol{y}_n \sim \text{Norm}(\boldsymbol{\mu}_{z_n}, \boldsymbol{\Sigma}_{z_n}) \qquad \text{for } n = 1, \ldots, N \tag{17}$$

Anglican code for this model is shown in Figure 4. Anglican provides stateful objects, which are referred to as random processes, to represent the predictive distributions for the cluster assignments $z$ and the observations $\boldsymbol{y}^k$ assigned to each cluster

$$z_{n+1} \sim p(\cdot \mid z_{1:n}, \alpha), \tag{18}$$
$$\boldsymbol{y}_{m+1}^k \sim p(\cdot \mid \boldsymbol{y}_{1:m}^k, \boldsymbol{\mu}_0, \kappa, \boldsymbol{\Psi}, \nu). \tag{19}$$

In this collapsed representation marginalization over the model parameters $\boldsymbol{\pi}$, $\boldsymbol{\mu}_{k=1:K}$, and $\boldsymbol{\Sigma}_{k=1:K}$ is performed analytically. Using the Iris dataset, a standard benchmark for mixture models that contains 150 labeled examples with 4 real-valued features, we optimize the marginal with respect to the subset of the parameters $\nu$ and $\alpha$ under uniform priors over a fixed interval. For this model, BOPP aims to maximize

$$p(\nu, \alpha | \boldsymbol{y}_{n=1:N}, \boldsymbol{\mu}_0, \kappa, \boldsymbol{\Psi})$$
$$= \iiiint p(\nu, \alpha, z_{n=1:N}, \boldsymbol{\pi}, \boldsymbol{\mu}_{k=1:K}, \boldsymbol{\Sigma}_{k=1:K} | \boldsymbol{y}_{n=1:N}, \mu_0, \kappa, \boldsymbol{\Psi}) \text{d}z_{n=1:N} \text{d}\boldsymbol{\pi} \text{d}\boldsymbol{\mu}_{k=1:K} \text{d}\boldsymbol{\Sigma}_{k=1:K}. \tag{20}$$

```
(defopt mvn-mixture [data mu0 kappa psi] [nu alpha]
  (let [[n d] (shape data)
        alpha (sample (uniform-continuous 0.01 100))
        nu (sample (uniform-continuous (- d 1) 100))
        obs-proc0 (mvn-niw mu0 kappa nu psi)]
    (loop [data data
           obs-procs {}
           mix-proc (dirichlet-discrete
                      (vec (repeat d alpha)))]
      (let [y (first data)]
        (if y
          (let [z (sample (produce comp-proc))
                obs-proc (get obs-procs z obs-proc0)
                obs-dist (produce obs-proc)]
            (observe obs-dist y)
            (recur (rest data)
                   (assoc obs-procs z (absorb obs-proc y))
              (absorb mix-proc z)))
          mix-proc)))))
```

Figure 1: Anglican query for hyperparameter optimization of a Gaussian mixture model, defined in terms of two parameters `nu` and `alpha`. A `mvn-niw` process is used to represent the marginal likelihood of observations under a Gaussian-inverse-Wishart prior, whereas a `dirichlet-discrete` process models the prior probability of cluster assignments under a Dirichlet-discrete prior. The command `produce` returns the predictive distribution for the next sample from a process. `absorb` conditions on the value of the next sample.

Figure 2: Bayesian optimization of hyperparameters in a Gaussian mixture model evaluated on the Iris dataset. Panels show the GP posterior as a function of number of evaluations, with the surface corresponding to the posterior mean and the color bars the posterior standard deviation. Optimization is performed over the parameter $\alpha$ of a 10-dimensional symmetric Dirichlet distribution and the degrees of freedom $\nu$ of the inverse-Wishart prior. At each evaluation we obtain an estimate of the log marginal $\log p(Y, \theta)$ obtained by performing sequential Monte Carlo inference with 1000 particles. The apparent maximum after initialization with 10 randomly sampled points lies at $\nu = 31$, $\alpha = 60$, and $\log p(Y, \theta) = -456.3$ (*left*). The surface after 10 optimization steps shows a new maximum at $\nu = 9.2$, $\alpha = 0.8$, and $\log p(Y, \theta) = -364.2$ (*middle*). After 40 steps and 50 total evaluations this optimum is refined to $\nu = 16$, $\alpha = 0.2$, and $\log p(Y, \theta) = -352.5$ (*right*).

Figure 2 shows GP regressions on the evidence after different numbers of the SMC evaluations have been performed on the model. This demonstrates how the GP surrogate used by BO builds up a model of the target, used to both estimate the expected value of $\log p(Y, \theta)$ for a particular $\theta$ and actively sample the $\theta$ at which to undertake inference.

### 3.3 Extended Kalman Filter for the Pickover Chaotic Attractor

We next consider the case of learning the dynamics parameters of a chaotic attractor. Chaotic attractors present an interesting case for tracking problems as, although their underlying dynamics are strictly deterministic with bounded trajectories, neighbouring trajectories diverge exponentially[1]. Therefore

| (a) 1 iteration $\theta = [-1.478, 0.855]^T$ | (b) 20 iterations $\theta = [-2.942, 1.550]^T$ | (c) 100 iterations $\theta = [-2.306, 1.249]^T$ | (d) Ground truth $\theta = [-2.3, 1.25]^T$ |

Figure 3: A series of trajectories for different parameters, demonstrating convergence to the true attractor. The colormap is based on the speed and curvature of the trajectory, with rendering done using the program Chaoscope [11].

regardless of the available precision, a trajectory cannot be indefinitely extrapolated to within a given accuracy and probabilistic methods such as the extended Kalman filter must be incorporated [8, 9]. From an empirical perspective, this forms a challenging optimization problem as the target transpires to be multi-modal, has variations at different length scales, and has local minima close to the global maximum.

Suppose we observe a noisy signal $y_t \in \mathbb{R}^K$, $t = 1, 2, \ldots, T$ in some $K$ dimensional observation space were each observation has a lower dimensional latent parameter $x_t \in \mathbb{R}^D$, $t = 1, 2, \ldots, T$ whose dynamics correspond to a chaotic attractor of known type, but with unknown parameters. Our aim will be to find the MMAP values for the dynamics parameters $\theta$, marginalizing out the latent states. The established parameters can then be used for forward simulation or tracking.

To carry out the required MMAP estimation, we apply BOPP to the extended Kalman smoother

$$x_1 \sim \mathcal{N}\left(\mu_1, \sigma_1 I\right) \tag{21}$$

$$x_t = A\left(x_{t-1}, \theta\right) + \delta_{t-1}, \qquad \delta_{t-1} \sim \mathcal{N}\left(0, \sigma_q I\right) \tag{22}$$

$$y_t = C x_t + \varepsilon_t, \qquad \varepsilon_t \sim \mathcal{N}\left(0, \sigma_y I\right) \tag{23}$$

where $I$ is the identity matrix, $C$ is a known $K \times D$ matrix, $\mu_1$ is the expected starting position, and $\sigma_1, \sigma_q$ and $\sigma_y$ are all scalars which are assumed to be known. The transition function $A\left(\cdot, \cdot\right)$

$$x_{t,1} = \sin\left(\beta x_{t-1,2}\right) - \cos\left(\frac{5x_{t-1,1}}{2}\right) x_{t-1,3} \tag{24a}$$

$$x_{t,2} = -\sin\left(\frac{3x_{t-1,1}}{2}\right) x_{t-1,3} - \cos\left(\eta x_{t-1,2}\right) \tag{24b}$$

$$x_{t,3} = \sin\left(x_{t-1,1}\right) \tag{24c}$$

corresponds to a Pickover attractor [10] with unknown parameters $\theta = \{\beta, \eta\}$ which we wish to optimize. Note that $\eta$ and $-\eta$ will give the same behaviour.

Synthetic data was generated for 500 time steps using the parameters of $\mu_1 = [-0.2149, -0.0177, 0.7630]^T$, $\sigma_1 = 0$, $\sigma_q = 0.01$, $\sigma_y = 0.2$, a fixed matrix $C$ where $K = 20$ and each column was randomly drawn from a symmetric Dirichlet distribution with parameter 0.1, and ground truth transition parameters of $\beta = -2.3$ and $\eta = 1.25$ (note that the true global optimum for finite data need not be exactly equal to this).

Inference was performed on this data using the same model and parameters, with the exceptions of $\theta$, $\mu_1$ and $\sigma_1$. The prior on $\theta$ was set to a uniform in over a bounded region such that

$$p\left(\beta, \eta\right) = \begin{cases} 1/18, & \text{if} -3 \leq \beta \leq 3 \cap 0 \leq \eta \leq 3 \\ 0, & \text{otherwise} \end{cases}. \tag{25}$$

The changes $\mu_1 = [0, 0, 0]$ and $\sigma_1 = 1$ were further made to reflect the starting point of the latent state being unknown. For this problem, BOPP aims to maximize

$$p(\beta, \eta | y_{t=1:T}) = \int p(\beta, \eta, x_{t=1:T} | y_{t=1:T}) \mathrm{d}x_{t=1:T}. \tag{26}$$

Inference on the transformed marginal query was carried out using SMC with 500 particles. Convergence results were shown in Figure 6 in the main paper, while Figure 3 shows the simulated attractors generated from the dynamics parameters output by various iterations of a particular run of BOPP.

Figure 4: Convergence for HMM in terms of the cumulative best $\log p\,(Y,\theta)$ (*left*) and distance to the "true" $\theta$ used in generating the data (*right*). Solid line shows median over 100 runs, whilst the shaded region the 25/75% quantiles. Note that for the distance to true $\theta$ was calculated by selecting which three states (out of the 5 generates) that were closest to the true parameters.

### 3.4 Hidden Markov Model with Unknown Number of States

We finally consider a hidden Markov model (HMM) with an unknown number of states. This example demonstrates how BOPP can be applied to models which conceptually have an unknown number of variables, by generating all possible variables that might be needed, but then leaving some variables unused for some execution traces. This avoids problems of varying base measures so that the MMAP problem is well defined and provides a function with a fixed number of inputs as required by the BO scheme. From the BO perspective, the target function is simply constant for variations in an unused variable.

HMMs are Markovian state space models with discrete latent variables. Each latent state $x_t \in \{1, \ldots, K\}, t = 1, \ldots, T$ is defined conditionally on $x_{t-1}$ through a set of discrete transition probabilities, whilst each output $y_t \in \mathbb{R}$ is considered to be generated i.i.d. given $x_t$. We consider the following HMM, in which the number of states $K$, is also a random variable:

$$K \sim \text{Discrete}\{1,2,3,4,5\} \tag{27}$$
$$T_k \sim \text{Dirichlet}\{1_{1:K}\}, \quad \forall k = 1, \ldots, K \tag{28}$$
$$\phi_k \sim \text{Uniform}[0,1], \quad \forall k = 1, \ldots, K \tag{29}$$
$$\mu_0 \leftarrow \min\{y_{1:T}\} \tag{30}$$
$$\mu_k \leftarrow \mu_{k-1} + \phi_k \cdot (\max\{y_{1:T}\} - \mu_{k-1}), \quad \forall k = 1, \ldots, K \tag{31}$$
$$x_1 \leftarrow 1 \tag{32}$$
$$x_t | x_{t-1} \sim \text{Discrete}\{T_{x_{t-1}}\} \tag{33}$$
$$y_t | x_t \sim \mathcal{N}(\mu(x_{t-1}), 0.2). \tag{34}$$

Our experiment is based on applying BOPP to the above model to do MMAP estimation with a single synthetic dataset, generated using $K = 3$, $\mu_1 = -1$, $\mu_2 = 0$, $\mu_3 = 4$, $T_1 = [0.9, 0.1, 0]$, $T_2 = [0.2, 0.75, 0.05]$ and $T_3 = [0.1, 0.2, 0.7]$.

We use BOPP to optimize both the number of states $K$ and the stick-breaking parameters $\phi_k$, with full inference performed on the other parameters. BOPP therefore aims to maximize

$$p(K, \phi_{k=1:5} | y_{t=1:T}) = \iint p(K, \phi_{k=1:5}, x_{t=1:T}, T_{k=1:K} | y_{t=1:T}) \mathrm{d}x_{t=1:T} \mathrm{d}T_{k=1:K}. \tag{35}$$

As with the chaotic Kalman filter example, we compare to two PMMH variants using the same code transformations. The results, given in Figure 4, again show that BOPP outperforms these PMMH alternatives.

## Footnotes

[1] It is beyond the scope of this paper to properly introduce chaotic systems. We refer the reader to Devaney [7] for an introduction.