[Reviews · NeurIPS 2016]

Reviewer 1

Summary

The paper introduces Bayesian optimisation for parameter optimisation to complement the inference of a probabilistic programming environment.

Qualitative Assessment

Technical quality: The method seems technically solid with one exception: I do not believe in collapsing the posterior GP inference results to a single GP (ca. line 204). Given HMC samples of the GP hyperparameters, the posterior predictive distribution is a *mixture of GPs*, not a sum. Clearly one cannot trivially reduce a mixture of Gaussians to a single Gaussian, and the same is true for GPs here. (Consider e.g. a bimodal distribution of means which clearly cannot be reduced to a single Gaussian.) Reduction to a single GP may work fine as an approximation, but it should be presented and analysed as one. The experimental evaluation presented in the paper is very limited. There are more details in the supplementary material which look much better. Novelty: The idea of applying Bayesian optimisation within probabilistic programming seems novel and potentially very useful. The implementation also introduces some improvements to the Bayesian optimisation method. Impact: The paper seems to make a significant step towards making general purpose probabilistic programming generally useful and usable. Clarity: The paper was well-written and in general easy to read. Some details of the experiments are unfortunately only in the supplement, but that seems difficult to avoid within the current page limit. Update after author feedback: Your response on the mixture of GPs issue looks good, raising the corresponding sub-score accordingly.

Confidence in this Review

2-Confident (read it all; understood it all reasonably well)


Reviewer 2

Summary

This paper presents a framework for maximum margin margin a posteriori (MMAP) inference in probabilistic programs which can be generally applicable to systems that employ a general-purpose programming language for model specification such as Church, WebPPL, and Anglican. The only requirement is that the probabilistic programming system (PPS) supports an inference method that returns a marginal likelihood estimate. In more detail, the paper focuses on providing point estimates of some latent variables while marginalizing the others, where the former is achieved through GP-based Bayesian optimization. There are a few challenges in order to provide such a ‘black-box’ approach for PPS and the paper focuses on a particular implementation using Anglican.

Qualitative Assessment

* Update after rebuttal * After reading the other reviewers' feedback, I am downgrading my scores on technical quality and clarity. It seems that the paper needs a bit of work to improve on this. [Technical quality, whether experimental methods are appropriate, proofs are sound, results are well analyzed] I believe the paper is technically sound as it builds upon a solid probabilistic programming system such as Anglican and upon a theoretically founded technique for optimization of expensive functions such as Bayesian optimization. The results comparing to other Bayesian optimization methods (figure 5) are quite impressive but more useful insights should be provided as to why this is the case. In addition, for the actual problem of interest (i.e. doing MMAP), the authors should provide an example as to how the proposed approach compares to a “hard-coded” solution in terms of time and quality of the solution. For example when performing hyper-parameter optimization in a graphical model while marginalizing all the other latent variables, or using a structured prediction problem. This will provide the reader with a better understanding of how useful the framework is. [Novelty / originality] The general idea of using Bayesian optimization for tuning parameters in machine learning models is (of course) not new with one of the most popular papers presented in reference [24]. However, the introduction of such techniques into PPS give rise to several difficulties, namely (a) dealing with problem-independent priors, (b) unbounded optimization and (c) implicit constraints. The paper seems to address these in an effective way. [Potential impact or usefulness] The paper addresses an important inference problem, that of providing point (MAP) estimates for some latent variables while marginalizing the others in probabilistic programs. This is very useful in settings such as hyper-parameter optimization. To my knowledge, there has not been previous work addressing this particular problem. [Clarity and presentation, explanations, language and grammar, figures, graphs, tables, proper references.] The paper is very well written, clearly motivated with an example and the contributions explained step-by-step.

Confidence in this Review

2-Confident (read it all; understood it all reasonably well)


Reviewer 3

Summary

The paper considers Bayesian optimization for probabilistic programs (PPs). The specific interest is in marginal maximum a posterior (MMAP) estimate for a subset of the parameters in the model defined by a PP. Large parts of the paper are devoted to describing the programmin workflow and the specific PP Anglican.

Qualitative Assessment

The paper presents a implementation of Bayesian optimization to an interesting problem. I liked the content of the paper in general but not the presentation and had some doubts on the methods (as detailed below). However, the dislike of presentation might be my personal preference. I'm not expert on Probabilistic Programming systems in broad sense (I'm confident in using STAN and few others). Hence, I find it very difficult to follow the presentation which relies on syntax for Anglican. I'm sorry but I don't fully understand the syntax in figures 2 and 3 and the long paragraphs of syntax style text - or at least I would need to first study the respective manual. For example in lines 119-120, I don't understand what "(sample:id dist)" means. However, the "intellectual" content in figures and text is syntax independent - and I think it should be in order to justify publication. For this reason I think authors should revise the paper and explain what they do in plain words and standard mathematics notations. For example, there is no need to use awkward "[:power-dis]" to denote distribution when authors could just write \theta \sim p(\theta). I think the paper does not read very well in general. Authors use very condensed sentences that are hard to understand. For example, I did not understand the following: - line 144: "...sampling tested \theta values..." What are "tested \theta values"? - The list in lines 143-151 is too vague at this point. Could you, at least, refer to later sections when talking about things not defined yet (e.g. line 148). - line 200: "...combination...", what kind of combination; sum, product, ...? - line 201: "...significantly higher weigth". What is significant? How do you give weight to a kernel; do you have a prior over them and you sample GP's with different kernels? - line 204: What does mean "...linearity of GPs is exploited to convert the produced samples into a single GP."? - line 226: "If points are observed... target for the GP..." what are the points referred here and how are they observed? And what is target for the GP? - In line 166 and later authors talk about "problem independent hyperprior" but they never define that. The black box nature of their algorithm is an essential novelty here and the hyperprior plays major role there. Hence, it should be carefully justified. Then a more fundamental question. In the Introduction and Discussion (and in few other places) authors talk about searching for marginal maximum a posterior estimate but as far as I've understood correctly they search for maximum marginal likelihood. In line 143 authors talk about "A code transformation allowing the marginal likelihood (ML) to be evaluated at a fixed \theta". This is also what seems to be possible with algorithm in Figure 3. I cannot find details on where the prior for \theta comes into play in this algorithm. This might be a problem of presentation that I criticized above and, if so, should be corrected. One specific point is line 208 where \hat{u}_m^* is defined to be the maximum expected posterior density of \theta. However, earlier in line 138 it was defined to be _log_ marginal _likelihood_. I am a bit skeptical that the idea presented in section 4.4. works in general setting. How do you justify the choice of r_{\infty} for example? Let me tell an example from a real problem I'm working with currently. I'm interested in finding the (real) maximum a posterior estimate for \theta \in \Re^2. The prior is really vague and likelihood is really narrow leading to a situation where the meaningful posterior mass would cover only about 1% of authors hypercube. More spefically, the hypercube (in log \theta space) would be [0 1] x [0 1] and the MAP is around [0.9 0.9] and log p([0.9 0.9]+\epsilon*[1 1]|Y) + 10 < log p([0.9 0.9]|Y) for all epsilon greater than 0.1. Hence, the posterior mass is practically negligible if the difference between log posteriors is greater than 10 and a naive BO algorithm failes here. Another question relates to the dimension of \theta. How large can it be? Few minor specific comments: - Figure 3 caption: what does "probability p(y)=y according to value y = (C, \theta)" mean? Moreover, "C" is introduced much later than its first appearance in text or in reference to Figure 3. - line 137: "\hat{u}_m^*" the same thing is denoted by \hat{w} in line 195 - line 138: "log marginal likelihood log p(\hat{\theta}_m^*|Y)"? isn't this log marginal posterior? - line 163: definition/explanation of q-cond comes much later so I was at first confused here. - the font size in figures 5 and 6 is too small To summarize. I think the paper puts too much emphasis on "coding" before assuring that the theory behind the method really works. The paper could make a good example if formulated so that it concentrates on the method.

Confidence in this Review

2-Confident (read it all; understood it all reasonably well)


Reviewer 4

Summary

The paper explores marginal maximum a posteriori estimation of probabilistic program variables by utilized Bayesian optimization.

Qualitative Assessment

1.My general impression is that the Introduction should be short and sweet, and the Experiments should be richer. 2.GP which is used in Bayesian Program Optimization as surrogate leads to high computational budgets, How to handle it? 3.How to distinguish between inference and optimization?

Confidence in this Review

2-Confident (read it all; understood it all reasonably well)


Reviewer 5

Summary

This paper proposes to use Bayesian optimization to solve the marginal MAP problem where we can only access to the black-box maximum likelihood estimation such as MCMC. The authors use the expected improvement with GP surrogate model. They propose several modification to the standard Bayesian optimization using EI: (1) they affine transform the variable space to the unit hypercube, the place a problem-independent hyper-priors on the GP hyperparameters. (2) They use a bump like mean function. (3) Utilizing the problem structure, they use annealed importance sampling to optimize the acquisition function, which make the final result satisfy some implicit constraints.

Qualitative Assessment

This paper targets at an interesting problem, which can utilize the potential power of the Bayesian optimization, but the authors fail to deliver the potential due to the following detailed comments. At the same time, the paper is not well written, containing lots of discontinuity, typos. Some high-level comments: (1) NIPS is a machine learning conference, the authors can formulate the problem more mathematically/abstractly instead they use a lot of terminologies in the probabilistic programing which make the paper really difficult to digest. Mathematically, it is to solve the MMAP problem, it’s easy to just formulate this way, and the readers can more clearly see its contribution. (2) The authors claim that the method can solve the noisy function evaluation such as MCMC, but they use EI as acquisition function. It’s widely known that EI is very bad at handling the noisy functional evaluation. See the authors’ own reference on the recent literature review by Ryan Adams’s group. (3) In my opinion, the affine transformation is not a big deal. The bump-like mean function goes away from stationary mean function, the authors need to compare it to some BO method based on non-stationary mean function, such as “INPUT WARPING FOR BAYESIAN OPTIMIZATION OF NON-STATIONARY FUNCTIONS”. (4) The annealed importance sampling to optimize the acquisition function is relative slow, especially the dimension is high. Can authors mention the speed of their algorithm more explicitly? (5) In the experiments, I’m quite worried about the figure 5. The benchmarks such as spearmint, SMA, TPE shows no progress on the Branin, Hartmann-6, SVM and LDA which is contradictory to the previous results in the literature. I suppose the y-axis is the simple regret, then spearmint should progress very fast on Hartmann6, see the figure 3 in “Predictive Entropy Search for Efficient Global Optimization of Black-box Functions”. And can authors just show the median and standard deviation of their own method instead of plotting every sample path out, it’s distracting. (6) the authors should compare to more recent BO methods, such as local penalization, UCB/EST, entropy search.

Confidence in this Review

2-Confident (read it all; understood it all reasonably well)